# A Snapshot Picture of the Fungal Composition of Bee Bread in Four Locations in Bulgaria, Differing in Anthropogenic Influence

**DOI:** 10.3390/jof7100845

**Published:** 2021-10-09

**Authors:** Svetoslav G. Dimov, Lyuben Zagorchev, Mihail Iliev, Tereza Dekova, Ralitza Ilieva, Meglena Kitanova, Dimitrina Georgieva-Miteva, Martin Dimitrov, Slavil Peykov

**Affiliations:** Faculty of Biology, Sofia University “St. Kliment Ohridski”, 8 Dragan Tzankov Blvd., 1164 Sofia, Bulgaria; lzagorchev@biofac.uni-sofia.bg (L.Z.); miliev1@biofac.uni-sofia.bg (M.I.); dekova@biofac.uni-sofia.bg (T.D.); ralitza_ilieva@biofac.uni-sofia.bg (R.I.); m.kitanova@uni-sofia.bg (M.K.); d.georgieva@biofac.uni-sofia.bg (D.G.-M.); martin_dimitrov@uni-sofia.bg (M.D.); spejkov@biofac.uni-sofia.bg (S.P.)

**Keywords:** bee bread, fungal microbiota, anthropogenic influence, NGS-based metagenomics

## Abstract

Information about the fungal composition of bee bread, and the fermentation processes to which the fungi contribute significantly, is rather scarce or fragmentary. In this study, we performed an NGS-based metagenomics snapshot picture study of the fungal composition of bee bread in four locations in Bulgaria during the most active honeybee foraging period at the end of June 2020. The sampling locations were chosen to differ significantly in climatic conditions, landscape, and anthropogenic pressure, and the Illumina 2 × 250 paired-end reads platform was used for amplicon metagenomics study of the ITS2 region. We found that some of the already reported canonical beneficial core fungal species were present within the studied samples. However, some fungal genera such as *Monilinia*, *Sclerotinia*, *Golovinomyces*, *Toxicocladosporium*, *Pseudopithomyces*, *Podosphaera* and *Septoriella* were reported for the first time among the dominant genera for a honeybee related product. Anthropogenic pressure negatively influences the fungal composition of the bee bread in two different ways—urban/industrial pressure affects the presence of pathogenic species, while agricultural pressure is reflected in a decrease of the ratio of the beneficial fungi.

## 1. Introduction

Bee bread is made by the bees from a mixture of collected pollen, plant nectar, and honey which are subjected to fermentation to raise the nutrient value of the pollen. This fermentation is accomplished by bacteria and fungi, originating from the bees’ saliva as well as from those present within the corbicular pollen. It provides easily digestible carbohydrates, proteins, fats, minerals, and vitamins for the bee larvae and the adult bees, but also acts as a functional food (feed), possessing antioxidant and anti-inflammatory properties [1,2,3]. Even though many studies in the last years have been focused on the bacterial composition of bee bread [4,5,6], those on fungal composition are rather scarce despite the publication of some of them dating from the ’80s [7]. An exception is the study of Tauber et al. [8]. Few exceptions [9,10] are motivated by the fact that bee bread represents a commercial interest presenting beneficial properties for human health [3].

Even though bee bread fungal microbiota is strongly influenced by seasonal and ecological factors, some “core” fungal genera have been identified within the pollen and bee bread—*Cladosporium*, *Mucor*, *Alternaria*, *Botrytis*, *Penicillium*, *Aspergillus* and *Rhizopus* [11,12]. Many of these have an ambiguous nature acting as probiotics [13] and/or shaping the bees’ health and fitness [14] but also being producers of mycotoxins [15,16]. However, available data suggest that their positive influence overcomes the negative one, at least for the beehives [13,17,18].

Because the microbial composition of the honeybees’ microbiomes, as well as those of the honeybee products, is strongly influenced by the landscape and anthropogenic pressure [19], we decided to investigate the fungal composition in bee bread at the end of June 2020 in four different locations in Bulgaria (Figure 1). This period was chosen as one of the most active concerning the honeybees’ foraging activities while the locations were selected because of the different degrees of anthropogenic pressure, as well as the landscape and the climatic features. In all the apiaries in this study European honeybees (*Apis mellifera macedonica*) [20] were reared.

## 2. Materials and Methods

### 2.1. Sampling Locations

Location 1: Sofia. This is the capital city of Bulgaria, located in a vast valley surrounded by mountains all around. The climate is typically continental. The constant population is over 1,600,000 inhabitants, and some more 200,000–300,000 people come for work on a daily basis. The apiary is located in the district of Obelya (Geographic coordinates: 42.744263 N, 23.264150 E; Elevation: 550 m). A highway, several busy boulevards, a thermal power plant, and many industrial enterprises are located within a 1–2 km range. Two of the capital’s biggest residential areas with 120,000 and 80,000 inhabitants respectively are also located within the foraging range of the bees. The agricultural activities are scarce and limited to the production for the personal use of fruits and vegetables in the local gardens. The anthropogenic pressure is estimated as very high because of the urbanization and the presence of many industrial enterprises.

Location 2: Dushantzi. The village is located within a small valley in a pre-mountain rural area in the mid-western part of the country (Geographic coordinates: 42.698124 N, 24.262389 E; Elevation: 720 m). The climate is continental. The number of residents is about 700. The agricultural activities within the foraging range of the bees are developed but not very intensive (some potato crops fields and sheep breeding). Many preserved natural areas and forests are also located within the 1–2 km zone. A big dam can be found at 1.5 km air distance. The closest industrial enterprise (a mining company) is located at 7 km air distance. The anthropogenic pressure is estimated as moderate.

Location 3: Kalina. The village is located at the most North-West part of the country in the large Danube plane (Geographic coordinates: 44.068890 N, 22.767405 N; Elevation: 105 m). The climate is continental. The number of the constant inhabitants is about 40. The agricultural activities are intensive with sunflower as the main crop in large fields. A small dam used for irrigation is located 100 m from the apiary. No industrial enterprises are located within a 12 km range. The anthropogenic pressure is estimated as high because of intensive agriculture.

Location 4: Momchilovtzi. The village is located in the southern part of Bulgaria in the middle of the Rhodopes mountain (Geographic coordinates: 41.657372 N, 24.773179 E; Elevation 1180 m). The climate is continental but with a strong influence from the Eastern part of the Mediterranean Sea which is at about 60 km distance. The number of the constant inhabitants is about 1050. The main agricultural activity is animal farming, mostly sheep breeding in the high-mountain pastures (several herds of a few hundred sheep). Because of the limitations of the terrain, crop growing is limited to potatoes and beans cultivation for personal use in small fields. No industrial enterprises are located within a radius of more than 30 km. The anthropogenic pressure is estimated as very low.

### 2.2. Bee Bread Sampling

About 200 mg (the content of 3 bee bread containing cells) of 2–4 days old (estimated by the beekeepers) bee bread was taken in sterile conditions from 3 randomly chosen hives in each apiary in the four different locations, and pooled together. Samples were put immediately at −10 °C and stored and transported to the laboratory at the same temperature within no more than 8 h. In the laboratory the samples were stored at −20 °C.

### 2.3. DNA Isolation

The sample pools from the four apiaries were mixed with an electric homogenizer and a sterile pestle, then 50 mg of bee bread mixture were used for DNA isolation with Quick-DNA™ Fecal/Soil Microbe Microprep Kit (Zymo Research, Cat. No. D6012, Irvine, CA, USA) according to the instruction manual. The quality of the extracted DNA was estimated by agarose electrophoresis in 1% gel in a TBE buffer system while the exact concentration was measured with a Quantus™ Fluorimeter (Promega, Madison, WI, USA). The obtained DNA concentrations varied between 45 and 72 ng/µL.

### 2.4. Metagenomic Analyses

#### 2.4.1. NGS-Sequencing

The DNA samples were shipped in dry ice to the Novogene Company Ltd. (Cambridge, UK). The sequencing was performed on the Illumina HiSeq 2 × 250 bp paired-end reads platform with 30 k tags per sample. The ITS2 region was sequenced with primers ITS3 (5′-GCATCGATGAAGAACGCAGC-3′) and ITS4 (5-TCCTCCGCTTATTGATATGC-3′), giving an amplicon product of 386 bp. The bioinformatics processing of the obtained data was executed by Novogene Company Ltd.

#### 2.4.2. Sequences Processing

The raw data were first processed for the removal of the barcodes and the primers’ sequences. The paired-end reads were merged by the FLASH V1.2.7 software tool [21]. The quality filtering of the raw tags was performed according to Bokulich [22] and Caporaso [23]. The tags were compared with the reference database (http://drive5.com/uchime/uchime_download.html, accessed on 24 September 2020) using the UCHIME algorithm [24] to detect chimera sequences, and then to obtain the effective tags. The chimera sequences were removed accordingly to Haas [25].

#### 2.4.3. Otus Analyses

All of the effective tags were analyzed by the Uparse v7.0.1001 software [26], and sequences with ≥97% similarity were assigned to the same OTUs. The representative sequences were analyzed by BLAST with Qiime v.1.7.0 [27], and against the Unite database [28]. MUSCLE v.3.8.31 [29,30] was used to construct the phylogenetic relationship of the OTUs representative sequences. OTUs abundance information was normalized using a standard sequence number corresponding to the sample with the fewest sequences. Subsequent analyses of alpha diversity were performed based on this output of normalized data. Krona charts were created using the Krona display [31].

#### 2.4.4. Alpha Diversity Analyses

The samples’ alpha diversity analyses were estimated based on the community richness indices by the Chao1 estimator (http://scikit-bio.org/docs/latest/generated/skbio.diversity.alpha.chao1.html#skbio.diversity.alpha.chao1, accessed on 3 September 2021) [32] and the ACE estimator (http://scikit-bio.org/docs/latest/generated/skbio.diversity.alpha.ace.html#skbio.diversity.alpha.ace, accessed on 17 October 2020) [33,34], as well the community diversity indices by the Shanon index (http://scikit-bio.org/docs/latest/generated/skbio.diversity.alpha.shannon.html#skbio.diversity.alpha.shannon, accessed on 17 October 2020) [35,36] and the Simpson index (http://scikit-bio.org/docs/latest/generated/skbio.diversity.alpha.simpson.html#skbio.diversity.alpha.simpson, accessed on 17 October 2020) [35,37]. The index of the sequencing depth was evaluated based on Good’s coverage (http://scikit-bio.org/docs/latest/generated/skbio.diversity.alpha.goods_coverage.html#skbio.diversity.alpha.goods_coverage, accessed on 18 October 2020) [38]. The index of phylogenetic diversity was calculated with the PD whole tree (http://scikit-bio.org/docs/latest/generated/skbio.diversity.alpha.faith_pd.html?highlight=pd#skbio.diversity.alpha.faith_pd, accessed on 18 October 2020) [39].

## 3. Results

### 3.1. NGS Sequencing Results

The analyses of the raw data are summarized in Table 1. The sequencing was of very good quality, allowing to proceed with the rest of the analyzes.

### 3.2. OTUs Counts

The results from the OTUs counts are displayed in Table 2. The numbers of the taxon tags varied between 504 and 3556 (on average 1610), while the number of the OTUs—between 156 and 281 (on average 198). The percentage of the unique tags varied between 0.84 % and 5.69 % (on average 2.77 %).

### 3.3. OTUs Taxonomic Annotation

The percentages based on the taxonomically annotated OTUs of the fungal phyla are presented in Table 3, while the Krona charts for the four locations in Figure 2, Figure 3, Figure 4 and Figure 5 display the relative abundances of the lower taxonomic ranks. The most abundant phylum was Ascomycota, accounting for more than 99 % of the OTUs found in all locations. Basidiomycota comes in second place, insignificantly represented at every location (0.04–0.30%), except for Dushantzi. Mucorcomycota was scarcely presented and only in Sofia with 0.30%. Of the fungal OTUs in Dushantzi, 0.06% were unclassified.

All dominant genera accounted for more than 1% of the overall genera in the four locations. These belonged to the Ascomycota phylum, and are the focus of this study. They are listed in Table 4 while their incidence in the four locations is shown on the Venn diagram in Figure 6.

### 3.4. Alpha Diversity Analyses

The calculated alpha diversity indexes are presented in Table 5.

## 4. Discussion

The NGS data reads used for this study was of good quality as illustrated by the values in Table 1, especially those of Q20, Q30, and the percentages of the effective tags, allowing correct further analyses. However, some confusion could arise as a result of the relatively high percentages of unclassified tags which varied approximately between 94% and 98%. A high percentage of unclassified fungi when ITS-based methods are employed is not unusual and could be explained by the incompleteness of the international databases with data of environmental samples [8,19,40]. For example, it varies from above 50% to 80% for environmental samples [40,41,42] to “significant” [43] and above 98% [44] for fermented foods and products.

Our results are close to those reported in one study of the corbicular pollen and bee bread, where the authors reported more than 93% Ascomycota OTUs counts [10]. Thus, the close percentages in the four locations, despite the differences in the taxon tags and the OTUs accounted for (Table 2), indicate that these numbers represent a tendency, and they are not random variations as a result of the chosen methodology.

Concerning the results of the annotation of the NGS data (Table 2), close numbers of the total tags and the unclassified tags were observed, except for Sofia where they were significantly lower. However, Sofia comes in first place in the percentage of the unique tags, despite being in third place in the number of taxon tags. Concerning the number of OTUs, the numbers were again close, except for Kalina where it was significantly higher. Nevertheless, Kalina comes in last place in the number of unique tags, and in second place in the number of the taxon tags. The highest number of the taxon tags was observed in Dushantzi, but it had one of the lowest OTUs number. The lowest numbers of the OTUs and the taxon tags were observed in Momchilovtzi, but it was second place in the number of the unique tags. All these observations led us to the conclusion that no correlations between the numbers of the total tags, the taxon tags, the OTUs, and the unique tags could be formulated, neither for the environmental conditions, nor the degree of the anthropogenic pressure, as well as the richness and the evenness of the fungal microbiota within the bee bread.

In practice, all fungi which could have an impact on bee bread fermentation and quality belonged to Ascomycota. Basidiomycota, Mucoromycota, and the unclassified phyla were represented only symbolically (Table 3, Figure 2, Figure 3, Figure 4 and Figure 5). This is the reason why the focus of our study was further put only on the Ascomycota dominant genera, represented by more than 1% percent of the annotated taxon tags within the samples from the four locations.

Members of 16 fungal genera appeared among the dominant species. Among the core fungal microbiota species reported in the literature [11,12], members of only three genera were found in our study—*Cladosporium*, *Penicillium*, and *Alternaria*. From them, only *Cladosporium* was present in the samples from all four locations, while *Penicillium* and *Alternaria* were present in the samples coming from three locations. Till now *Monilinia* was not reported to be part of the core fungal microbiota species. However, members of the genus were found within the samples of two of the locations. Members of all the other dominant genera were accounted for only one location.

Members of the *Cladosporium* genus come in first place by average combined content of 54% in all the locations, and they are also in first place in the samples from Dushantzi, Momchilovtzi and Sofia, being accounted for with more than 60% of all of the OTUs counts. The only location where *Cladosporium* is in second place is Kalina (15% of the OTUs counts), where the first place was occupied by members of the *Toxicocladosporium* genus (24% of the OTUs counts). This filamentous fungus has been reported previously to be one of the predominant genera within corbicular pollen and bee bread [9,10,11,13]. Its predominance within the core microbiota could be explained by its ability to secrete organic acids which preserve the collected pollen, by the secretion of extracellular enzymes which raise the nutritional value of the pollen [13], as well as by inhibiting pathogenic bacteria and fungi, for example, *Ascoshpaera apis* [10].

Members of the *Penicillium* genus were found at three of the locations (Momchilovtzi, Sofia, and Kalina) with percentages from 1% to 8% of the OTUs counts (average content of 3% among the whole study), but even higher contents have been reported [9]. They have been reported to be present in honey [45,46], in bee pollen [11], and even in stingless bee products [47]. This could be easily explained by their important role [48], similar to that of the *Cladosporium* species.

*Alternaria* species were also found in three of the four locations, being absent only in Kalina, and accounted for between 1% and 4%, and a total of 2% in the whole study. Although some species are phytopathogenic [49], they are also placed among the core members of the bees and bee products microbiotas [11,45,46]. They are important species, playing roles within the beehives, similar to those of *Cladosporium* and *Penicillium* species. Their widespread occurrence could also be supported by the recent findings that they possess the potential to inhibit the growth of the causative agents of the American foulbrood disease (AFB) and chalkbrood disease (CBD) [18].

*Monilinia* was also found within the samples in more than one location—Dushantzi and Sofia with 9% and 2% of the OTUs counts respectively. Almost no information about the presence of these species within bee products is published, except that bees could be vectors for the propagation of the molds belonging to this genus [50]. Evidently, *Monilinia* species possess the ability to thrive within fermenting bee bread. However, it was never reported to be found among the dominant fungal species in two of the locations. This unusual finding allows us to speculate that even though the genus *Monilinia* is not a member of the core fungal genera, it might play some role in the fermentation of bee bread. This guess is further supported by its third place with 9% of the OTUs counts in the sample from Dushantzi.

All other dominant fungal genera were found in only one location, despite being or not being members of the core genera.

Dushantzi was the only location where the *Botrytis* genus was present among the dominant genera, being at second place by the number of OUTs accounted for. It was also the only single dominant genus in addition to those already discussed. Members of this genus are plant pathogens. However, they are also members of the core bees-related genera. Spores of the genus have been found in pollen [11] and honey [46], but they have also been reported to have a beneficial effect on the development of worker bees [51], and also have a nutritional value [13].

Similar to Dushantzi, within the sample from Momchilovtzi only one genus was present among the dominant ones in addition to the other three core genera—in this case, *Sclerotinia*, which accounted for 34% of the OTUs. This is a surprising finding, keeping in mind that members of this genus are plant pathogens [52], as well as considering the extremely high percentage of the occurrence which cannot be considered as random or as a flaw of the methodology. One can only speculate that this fungus could have a similar impact on the bees and bee products as the other already discussed plant pathogens. However, this hypothesis should be debated with great caution because of the lack of supporting scientific data, except for one report of the association of *Sclerotinia* sp. with stingless bees [53].

Within the samples from Sofia, in addition to the three core genera, *Cladosporium*, *Penicillium* and *Alternaria*, two additional dominant genera were accounted for—*Ascosphaera* and *Golovinomyces*. *Ascoshaera* is a rigorous bee pathogen, being the causative agent of CBD, so being in second place by 34% of the annotated OTUs was an unexpected finding. More strikingly, when the samples were taken, the beekeeper selected hives that in his opinion were healthy. One possible explanation of the observed paradox is the ability of *Alternaria* and some lactic acid bacteria to inhibit the growth of this pathogen [18,54]. Members of the *Golovinomyces* are strict plant pathogens [55], and, until now, they have never been reported to be associated with bees and bee products. Once again, some plausible explanations of this finding are, if they are not some kind of contamination, they could have some nutritional value—either by themselves, or by secreting extracellular enzymes raising the nutritional value of the pollen.

The greatest number of genera among the dominant ones was found in the samples from Kalina—10, from which only two were among the core genera. In addition to *Cladosporium* and *Penicillium* (15% and 8% of the annotated OTUs counts respectively), *Toxicocladosporium*, *Pseudopithomyces*, *Camarosporium*, *Paraconiothyrium*, *Podosphaera*, *Paraphaeosphaeria*, *Periconia* and *Septoriella* were accounted for. The picture is even more astounding because the first and the second place by the number of annotated OTUs were occupied by non-core genera *Toxicocladosporium* and *Pseudopithomyces* with 24% and 18% respectively, while the third place with 15% is claimed by the core *Cladosporium* and the non-core *Camarosporium* genera. The remaining non-core dominant genera were accounted for with only 1% to 4%.

Until now, *Toxicocladosporium* genus has never been reported to be associated with bee products and bees. Members of the genus were found in many different environments such as soils [56], marine waters [57], household air conditioners [58], bronchoalveolar lavage fluid [59], different plants [60,61], and even in insects—in the planthopper gut [62] and the *Thitarodes* larvae hemolymph [63]. However, members of this genus have never been established as strict pathogens of the plant and animal species where they thrive. The high percentage of the *Toxicocladosporium* among the annotated OTUs, placing it at first place, is difficult to explain. One plausible hypothesis is that members of this genus are not pathogens, and at the same time are closely related to the other *Cladosporiaceae* genus observed, which is the most common core genus *Cladosporium*. Thus, it could play a similar role in the bee bread fermentation process. Still, further omics analyses are needed to confirm or reject this hypothesis.

Among the dominant genera in Kalina three members of the *Didymosphaeriaceae* family were found—*Pseudopithomyces*, which arrives at second place with 18% of the annotated OTUs, and *Paraconiothyruium* and *Paraphaeosphaeria* with 4% and 1% respectively. *Pseudopithomyces* species have been reported to be present in different ecological niches—freshwaters [64], seawaters [65], where they could also be associated with algae [66], to be endophytes [67] and/or plant pathogens [68], and to be present within the human fungal microbiome, together with *Paraphaeosphaeria* species [69], but most importantly they were found to be part of a fruit fly species microbiome [70]. Most likely, with 18% of the annotated OTUs counts their presence is not an accident, a hypothesis supported by the fact that they have been found in human and insect microbiomes. *Paraconithyrium* species have already been reported for corbicular pollen [9], while they are mostly endophytes and bioremediators [71,72]. They were also declared safe for honeybees as biocontrol agents used for soil treatments in the USA and the EU [73]. Being present within the corbicular pollen indicates that maybe they are also present within the bees’ microbiome, and to be transferred to bee bread from the insects’ saliva used for the construction of pollen granules. As for their 4% of the annotated OTUs counts, they have the potential to have a noticeable impact on bee bread fermentation. *Paraphaeosphaeria* species are also usually plant endophytes [74]. However, finding it within bee bread is not a surprising fact, being already reported to be present within honey samples [75], as well as to be reported as a gut endosymbiont for other insect species [76].

Within the Kalina samples, with 15% of the annotated OTUs counts, members of the *Camarosporium* genus disputed the third place of occurrence with those of the core genus *Cladosporium*, showing, therefore, that this finding is not a random chance. Information about this genus is rather scarce, its members being mostly known to be saprobes or plant pathogens [77]. However, further investigations are needed because they have been reported to be present within honey samples [78].

The last dominant genera found in Kalina bee bread samples were *Podosphaera*, *Periconia* and *Septoriella* accounted for by 1% of the annotated OTUs counts. From them, only members of the *Periconia* genus were reported to be found in honeybees-related products [78], and in an animal in general, a *Haliconia* sp. sponge [66]. The other two are plant pathogens [79,80]. Despite being among the dominant genera, the impact of these three would be minimal, if not non-existent, because of their very low content, especially for those which have never been associated with bees or insects. Additionally, *Periconia* sp. was recently reported as a human pathogen causing corneal ulcers [81].

If we compare the dominant genera composition of the four locations (Figure 6) regarding the landscape and the anthropogenic influence, the lowest number of dominant genera, in total 4, is observed in Dushantzi and Momchilovtzi, where the landscape is hilly and mountainous, and the anthropogenic pressure is minimal or not existent. On the other hand, the highest numbers of dominant genera were observed in Sofia and Kalina, respectively 6 and 10, where the anthropogenic pressure is high, and the landscape is flat, and where plant, human, and honeybee pathogens were accounted for. However, two types of anthropogenic pressure can be differentiated—urban/industrial and agricultural. The agricultural pressure reflects on the overall diminution of the beneficial fungal genera—only two core genera were present in Kalina with a combined percentage of 23%, as well as on the increase of the number of the non-canonical fungal genera, mainly by plant pathogens and saprophytes. On the other hand, the urban and industrial anthropogenic pressure did not reflect on the combined percentage of the beneficial fungal genera—at the three locations three core genera were observed with combined percentages from 61% to 71%, surprisingly the highest being observed in Sofia where the apiary was contaminated with *Ascosphaera* but showed no symptoms. This observation coincides with the well-documented ability of the honeybees to adapt to urban areas [82,83]. Nevertheless, the pressure of the urban conditions does have some negative impact which in our case was the observation of pathogenic fungi.

As the methodology used in the study comprises a PCR step of amplification of a part of the fungal ITS2 regions, some concerns about the validity of the data could arise. Despite this methodology being imposed itself globally as a gold standard for metagenomic studies, and research of its validity having been published [84], the alpha diversity indexes should be examined in every case (Table 5). The calculated values of the Chao1 and the ACE indices were close to the observed numbers of the fungal species, which is not a surprise because the calculated Good’s coverage index in all samples was 1.000, meaning that all samples were representative accounting for all present species within the samples [33,38,85]. Finally, Faith’s phylogenetic diversity PD whole tree indexes were calculated, which account for the phylogenetic realization of the species richness but do not consider the species abundance [39]. Keeping these considerations in mind, the results presented in this study should be considered as correct and informative, despite the fact that some minor deviances in the cited percentages of the annotated OTUs tags could not be completely excluded.

## 5. Conclusions

A bee bread microbiota survey study was performed in locations with different landscapes, climatic conditions, and degrees of anthropogenic pressure within a narrow timeframe of 8 days in the most active honeybees foraging period at the end of June. Some of the core honeybee-related fungal genera which were reported in the scientific literature were found, some others were reported to our knowledge for the first time within the dominant genera: *Monilinia*, *Sclerotinia*, *Golovinomyces*, *Toxicocladosporium*, *Pseudopithomyces*, *Podosphaera* and *Septoriella*. We also found that the degree of the anthropogenic pressure does have a negative impact on the bee bread fungal microbiota, the strongest one being that of intensive agriculture.

## Figures and Tables

**Figure 1 jof-07-00845-f001:**
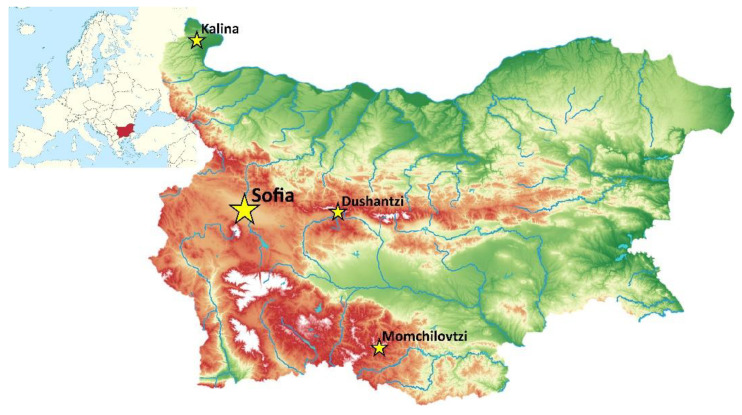
Locations of the apiaries from which the samples were taken.

**Figure 2 jof-07-00845-f002:**
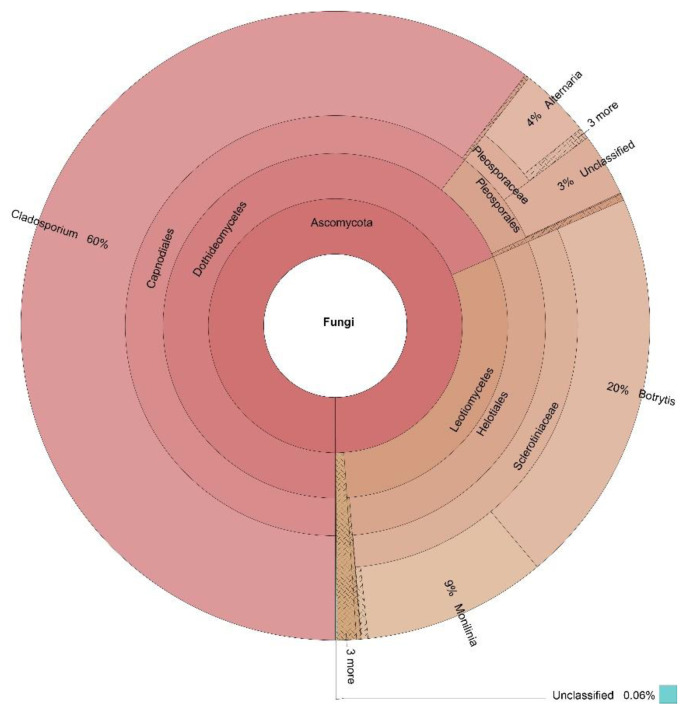
Krona display of the fungal composition of the samples from Dushantzi.

**Figure 3 jof-07-00845-f003:**
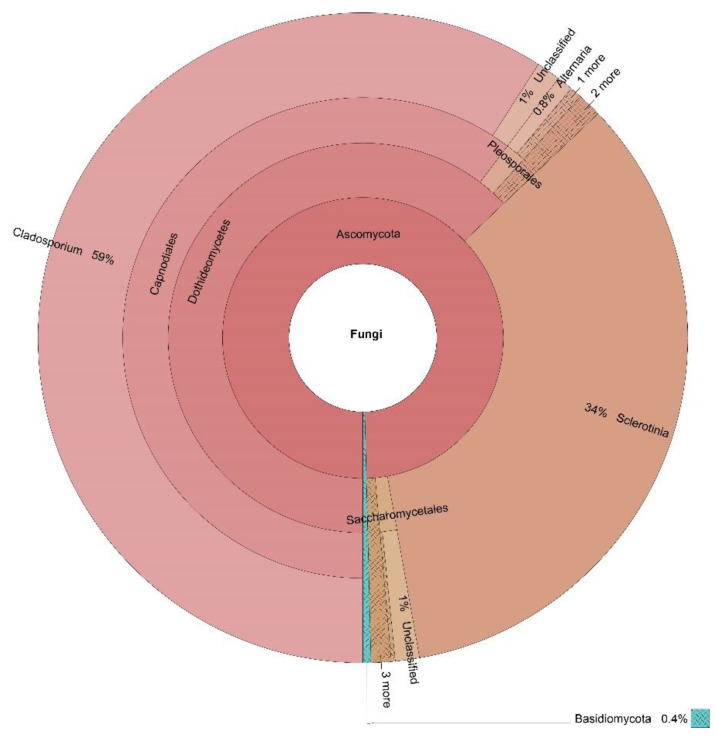
Krona display of the fungal composition of the samples from Momchilovtzi.

**Figure 4 jof-07-00845-f004:**
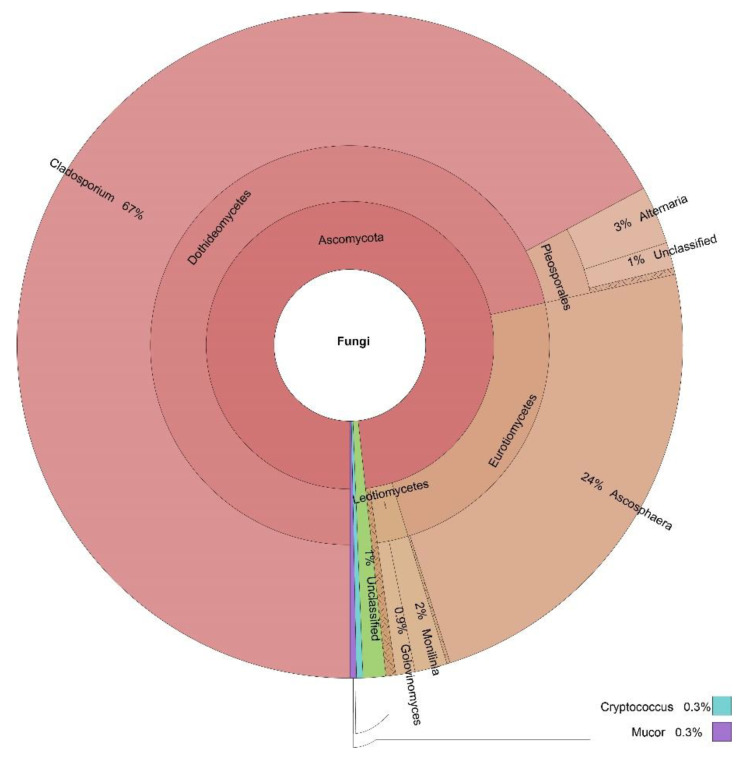
Krona display of the fungal composition of the samples from Sofia.

**Figure 5 jof-07-00845-f005:**
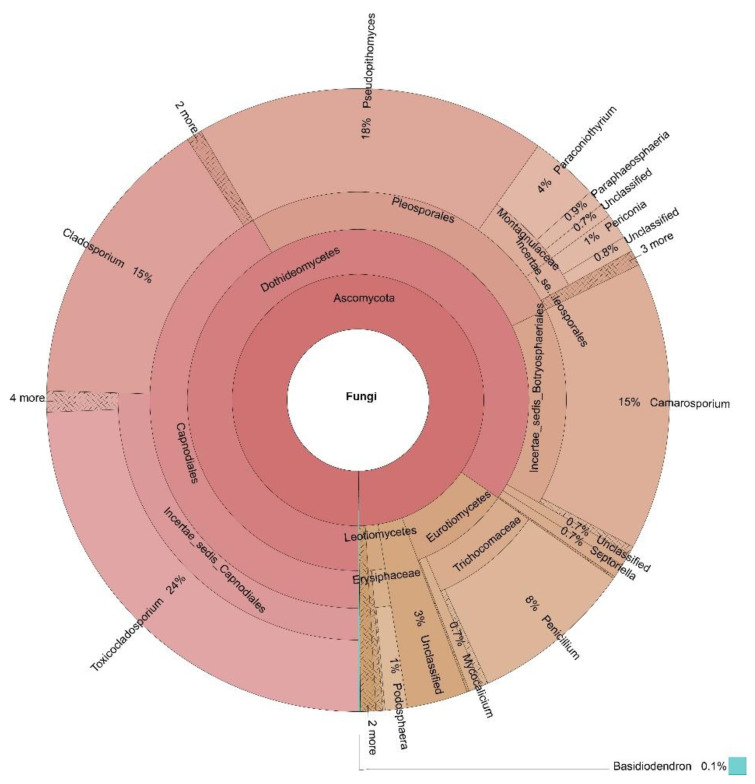
Krona display of the fungal composition of the samples from Kalina.

**Figure 6 jof-07-00845-f006:**
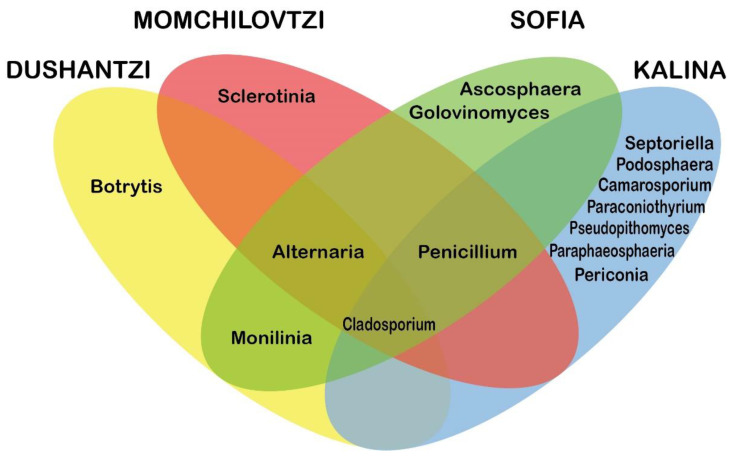
Venn diagram of the dominant fungal genera in the four locations.

**Table 1 jof-07-00845-t001:** Results from the NGS sequencing of the samples.

Location	Number of the Raw Paired-End Reads	Number of the Raw Tags	Number of the Clean Tags	Number of the Effective Tags	Number of Bases of the Effective Tags (Nt)	Average Length of the Effective Tags (Nt)	Q20 Value	Q30 Value	GC Content Percentage in Effective Tags	Percentage of Effective Tags in Raw Paired End
Dushantzi	139,921	138,997	138,469	137,837	50,312,892	365	99.39	97.57	54.30	98.51
Momchilovtzi	116,925	116,125	115,783	115,635	41,254,729	357	99.49	97.95	58.24	98.90
Sofia	86,102	84,754	83,876	83,679	30,482,594	364	98.24	94.94	59.32	97.19
Kalina	112,607	108,655	102,645	102,273	37,731,509	369	97.10	92.14	62.91	90.82

**Table 2 jof-07-00845-t002:** Summarization of the annotation of the NGS data.

Location	Total Tags	Unclassified Tags	Taxon Tags	% of the Unclassified Tags	Unique Tags	% of the Unique Tags	OTUs
Dushantzi	275,674	267,432	3556	97.01%	4686	1.70%	169
Momchilovtzi	231,270	224,174	504	96.93%	6592	2.85%	156
Sofia	167,358	157,204	638	93.93%	9516	5.69%	185
Kalina	204,546	201,080	1742	98.31%	1724	0.84%	281

**Table 3 jof-07-00845-t003:** Fungal phyla found in the four locations.

	Dushantzi	Momchilovtzi	Sofia	Kalina
Ascomycota	99.94%	99.96%	99.40%	99.90%
Basidiomycota	-	0.04%	0.30%	0.10%
Mucoromycota	-	-	0.30%	-
Unclassified	0.06%	-	-	-

**Table 4 jof-07-00845-t004:** Relative abundance of the Ascomycota genera found in the four locations ^1^.

	Dushantzi	Momchilovtzi	Sofia	Kalina	Average
Cladosporium	60%	59%	67%	15%	54%
Penicillium	0%	1%	1%	8%	3%
Alternaria	4%	1%	3%	0%	2%
Monilinia	9%	0%	2%	0%	3%
Sclerotinia	0%	34%	0%	0%	9%
Ascosphaera	0%	0%	24%	0%	6%
Toxicocladosporium	0%	0%	0%	24%	6%
Botrytis	20%	0%	0%	0%	5%
Pseudopithomyces	0%	0%	0%	18%	5%
Camarosporium	0%	0%	0%	15%	4%
Paraconiothyrium	0%	0%	0%	4%	1%
Podosphaera	0%	0%	0%	1%	0%
Golovinomyces	0%	0%	1%	0%	0%
Paraphaeosphaeria	0%	0%	0%	1%	0%
Periconia	0%	0%	0%	1%	0%
Septoriella	0%	0%	0%	1%	0%

^1^ The sum of the percentages does not equal 100% because of the presence of unclassified tags.

**Table 5 jof-07-00845-t005:** Alpha diversity indexes.

Location	Observed Species	Shannon	Simpson	Chao1	ACE	Goods Coverage	PD Whole Tree
Dushantzi	169	2.140	0.635	180.538	183.600	1.000	14.665
Momchilovtzi	156	2.963	0.727	171.120	177.229	1.000	13.575
Sofia	185	2.924	0.795	200.120	204.572	1.000	12.889
Kalina	281	2.178	0.506	281.000	281.000	1.000	17.992

## Data Availability

Not applicable.

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
