# Peer review of "A Snapshot Picture of the Fungal Composition of Bee Bread in Four Locations in Bulgaria, Differing in Anthropogenic Influence"

_jof, 2021, doi:10.3390/jof7100845_

Round 1

Reviewer 1 Report

The manuscript has a scientific value and novelty, is written legibly and all information are presented in a logical way. The introduction describes an actual state of the art in this field. The results are presented in tables and figures, and the Authors compared the obtained results to those from the literature. However, there are some issues that authors should refer in Introduction, Results, Discussion and Material and Methods.

Comments to Authors:

Introduction

Bee bread is a basic food from the bee family and plays an essential role in the production of other bee products. Therefore, there should be added some data about the about the role of fungi in Apis mellifera development as in https://onlinelibrary.wiley.com/doi/10.1002/yea.3665

Moreover, the impact of anthropogenic landscapes on honey bees should be more emphasis. Add some sentences about the impact of fungi on bees in anthropogenic landscapes. To support this sentence cite i.e. https://doi.org/10.3390/pathogens10030381

Materials and Methods

The main concern is about the number of checked samples.

How many samples of beebread was checked?  One from each locality? In total: 4 samples?

Line 97 -99

About 200 mg (about the content of 3 randomly chosen bee bread containing cells)… from 3 randomly chosen hives in 4 locations

What was one sample : a bee bread from 3 hives pooled together? – from one locality there was only one sample – if yes it is far too small to generate any data

Lines 102-107

The samples of bee bread from 2.2. should be mixed, powder and then 50 mg of such powder should be taken to DNA isolation

What was the methodology of DNA isolation? DNA was isolated  from one pooled  sample or from one bee bread containing cell?

Researches should repeated their study if there are only 4 samples of beebread studied.

Lines 197-201

Authors should discuss their data with the latest papers from that filed. To support this sentence cite   https://onlinelibrary.wiley.com/doi/10.1002/yea.3665 and https://doi.org/10.3390/pathogens10030381

Author Response

Dear reviewer,

First, on behalf of all authors I would like to thank you for your valuable contribution to our manuscript!

Please, find below our responses to your remarks:

Comments and Suggestions for Authors

The manuscript has a scientific value and novelty, is written legibly and all information are presented in a logical way. The introduction describes an actual state of the art in this field. The results are presented in tables and figures, and the Authors compared the obtained results to those from the literature. However, there are some issues that authors should refer in Introduction, Results, Discussion and Material and Methods.

  • We are very grateful for the esteem and evaluation of our work!

Comments to Authors:

Introduction

Bee bread is a basic food from the bee family and plays an essential role in the production of other bee products. Therefore, there should be added some data about the about the role of fungi in Apis mellifera development as in https://onlinelibrary.wiley.com/doi/10.1002/yea.3665

  • We are grateful for the suggestion, and included the citation.

Moreover, the impact of anthropogenic landscapes on honey bees should be more emphasis. Add some sentences about the impact of fungi on bees in anthropogenic landscapes. To support this sentence cite i.e. https://doi.org/10.3390/pathogens10030381

  • We are grateful for the suggestion, and included the work

Materials and Methods

The main concern is about the number of checked samples.

How many samples of beebread was checked?  One from each locality? In total: 4 samples?

  • We collected 3 samples from 3 different beehives in each apiary in the four locations which were combined in order to have a more representative overall picture.

Line 97 -99

About 200 mg (about the content of 3 randomly chosen bee bread containing cells)… from 3 randomly chosen hives in 4 locations

What was one sample : a bee bread from 3 hives pooled together? – from one locality there was only one sample – if yes it is far too small to generate any data

  • The samples from the 3 different hives in each apiary were pooled together. We made the appropriate changes within the text.

Lines 102-107

The samples of bee bread from 2.2. should be mixed, powder and then 50 mg of such powder should be taken to DNA isolation

What was the methodology of DNA isolation? DNA was isolated  from one pooled  sample or from one bee bread containing cell?

Researches should repeated their study if there are only 4 samples of beebread studied.

  • In total 12 samples (3 samples from the four locations) were studied. We clarified this section of the text as suggested.

Lines 197-201

Authors should discuss their data with the latest papers from that filed. To support this sentence cite   https://onlinelibrary.wiley.com/doi/10.1002/yea.3665 and https://doi.org/10.3390/pathogens10030381

  • We are very grateful for this suggestion and included the citations.

Reviewer 2 Report

Manuscript deal with fungal composition of the bee bread sampled at different locations. Generaly, it is interesting but could be improved if there is some link with bacterial composition and/or probiotic characteristic of fermented pollen samples.

Minor:

l 17 delete word Background

l 21 delete word (2) Methods:

l 23 delete (3) Results

l 27 delete (4) Conclusions:

l 38 ...bee larvae... 

Explain in which context imagoes are mentioned here (that life stage is not taking the food)

l 57 latin names must be written in italic

use word reared instead of cultivated

l 90 ...domestic animal..?

figures and tables please insert at first possible place after first mentioning

list of references must be edited, e.g. names of journals must be used as abbreviations

Author Response

Dear reviewer,

First, on behalf of all authors I would like to thank you for your valuable contribution to our manuscript!

Please, find below our responses to your remarks:

Comments and Suggestions for Authors

Manuscript deal with fungal composition of the bee bread sampled at different locations. Generaly, it is interesting but could be improved if there is some link with bacterial composition and/or probiotic characteristic of fermented pollen samples.

  • We are very grateful for this suggestion. However, this work is a part of a bigger project which aims to monitor the bacterial and fungal composition during two years period. We will make a broader article when the full experimental work is done, and we will try to make such correlations.

Minor:

l 17 delete word Background

l 21 delete word (2) Methods:

l 23 delete (3) Results

l 27 delete (4) Conclusions:

  • We deleted the sections titles. We are grateful for this remark!

l 38 ...bee larvae... 

Explain in which context imagoes are mentioned here (that life stage is not taking the food)

  • We mentioned the imagoes because bee bread is consumed also by the adult bees, especially in winter when fresh pollen is not available ( https://doi.org/10.2478/jas-2013-0025, https://doi.org/10.1016/j.tifs.2019.08.021 )

l 57 latin names must be written in italic

use word reared instead of cultivated

  • We are very grateful for showing us this mistake, as well as for the suggestion!

l 90 ...domestic animal..?

  • We included a small text accordingly.

figures and tables please insert at first possible place after first mentioning

list of references must be edited, e.g. names of journals must be used as abbreviations

  • They were included as required by the “Instructions to the authors”. For the references we used the MDPI Endnote style.

Round 2

Reviewer 1 Report

All my concerns were addressed.  I don't have further suggestions or corrections.